# Quasi-normal modes of microscopic Black Holes in non-perturbative quantum gravity

Alexey S. Koshelev,[1,2,*] Chenxuan Li,[1,†] and Anna Tokareva[3,4,‡]

[1]*School of Physical Science and Technology,*

*ShanghaiTech University, 201210 Shanghai, China*

[2]*Departamento de Física, Centro de Matemática e Aplicaçoes (CMA-UBI),*

*Universidade da Beira Interior, 6200 Covilhã, Portugal*

[3]*School of Fundamental Physics and Mathematical Sciences,*

*Hangzhou Institute for Advanced Study, UCAS, Hangzhou 310024, China*

[4]*International Centre for Theoretical Physics Asia-Pacific, Beijing/Hangzhou, China*

## Abstract

Non-pertrubative quantum gravity formulated as a unitary four-dimensional theory suggests that certain amount of non-locality, such as infinite-derivative operators, can be present in the action, in both cases of Analytic Infinite Derivative gravity and Asymptotically Safe gravity. Such operators lead to the emergence of Background Induced States on top of any background deviating from the flat spacetime. Quasi-normal modes (QNMs) corresponding to these excitations are analyzed in the present paper with the use of an example of a static nearly Schwarzschild black hole. We target micro-Black Holes, given that they are strongly affected by the details of UV completion for gravity, while real astrophysical black holes can be well described in EFT framework. We find that frequencies of QNMs are deviating from those in a General Relativity setup and, moreover, that the unstable QNMs are also possible. This leads to the necessity of constraints on gravity modifications and lower bounds on masses of the stable micro-Black Holes.

* askoshelev@shanghaitech.edu.cn

† lichx4@shanghaitech.edu.cn

‡ tokareva@ucas.ac.cn

## I. INTRODUCTION

Black holes (BHs) were predicted by Einstein's General Relativity (GR) [1] and are widely studied since the pioneering findings by Schwarzschild [2]. Although known solutions such as Schwarzschild BH, Kerr BH [3] and many others describe isolated BHs in an empty space, real BHs live in a more complex world and are usually in a perturbed state. BH perturbations manifest themselves by emitting energy in the form gravitational radiation, i.e. Gravitational Waves (GWs). This provides an exceptional opportunity to study the structure of BH *and* probe modified gravity theories for consistency.

Black hole perturbations can be studied either by systematically perturbing the BH metric and thus coming to the Regge-Wheeler equations [4–6] or by adding fields to the BH background and studying corresponding field perturbations (see [7, 8] for a review). The latter approach provides an easier insight into the problem by capturing the most essential effects of the perturbation analysis. For the linear perturbations, one can safely disregard a field back-reaction, and the essence of the study reduces to tackling a Klein-Gordon equation in a BH metric.

One of the most important classes of BH perturbations is the so-called quasi-normal modes (QNMs), which correspond to the dumped outgoing GWs which a perturbed BH emits at a ring-down stage. Obviously, the lower the dumping factor is the more long-range effect such perturbations have. Thus it is therefore very tantalizing to find out such modes in a spectrum of perturbations. Given the importance and especially our current abilities to observe QNMs due to incredible improvements in observational techniques in the recent times [9–13], QNMs is a subject of very intensive current study in different settings [14–47] as well as references therein and multiple other references.

While most studies of QNMs are performed in the GR framework, it is well-known that GR must be modified at high energies. Gravity modification will evidently change Einstein equations, but also, evidently, these changes should become relevant only at the energies around Planck scales (or at minimum above the scale of inflation [48]). Thus, they seem to be irrelevant for real astrophysical BHs in regard to their classical perturbations [49]. As it was demonstrated in [49] effects of gravity modifications on astrophysical BHs can be suppressed by incredibly small factors like $10^{-50} \div 10^{-162}$. Thus, at the corresponding energy scales, to a good approximation, perturbations can be studied in the context of effective field

theory (EFT) of GR[1] [49–58]. This picture, however, is different for micro-BHs ($\mu$BHs) which are discussed in various contexts [59–70]. $\mu$BHs can easily be contemplated to be of a mass close to the Planck scale and, therefore, they are sensitive to the details of UV completion for gravity.

In this paper, we turn our attention to studying QNMs in connection with gravity modifications, which are considered relevant for efforts of constructing non-perturbative UV complete theory of gravity. We therefore study QNMs of $\mu$Black Holes as they are strongly affected by the details of UV completion for gravity. Especially we pay a special attention to a peculiar properties of higher derivative operators which generate the so-called Background Induced States (BISs) [71–73] around any non-flat background. These new excitations emerge due to the presence of an infinite tower of derivatives on any curved background different from the Minkowski space-time. In fact they equally appear in a simple scalar field setup like for instance in String Field Theory based models and such modes are known not to spoil unitarity [74, 75]. These new modes can in particular appear in pairs with complex conjugate masses squared resulting in totally new effects. Such a specific issue requires special attention and will be discussed in detail in the due course hereafter.

We found the present work on the QFT paradigm in four-dimensional space-time and very important notions of renormalizability and unitarity are crucial guiding principles for this study. Very promising approaches in this direction are Analytic Infinite Derivative (AID) gravity [48, 76–79] and quantum effective action in asymptotically safe gravity [80–87] in its form-factor formulation [88, 89]. Both approaches contain higher and infinite derivative operators in the action. These operators are introduced following the arguments from somewhat different perspectives but their presence seems to be an inevitable property of a UV complete quantum gravity if it can be a four-dimensional field theory. BHs and their properties were intensively studied both in AID gravity [90–95] and asymptotically safe gravity [96–100]. While stability of BHs and corresponding QNMs were surely also studied before including the just cited papers, previously authors exercised a standard analysis without accounting newly emerged excitations, or BISs, mentioned above.

Before going to the main questions of the present paper, we emphasize that the infinite derivative operators lead to several other new obstacles and puzzles besides BISs. Even a question of substituting a given metric into equations of motion is an unfeasible proce-

---

[1] Here we assume that there are no new degrees of freedom apart from two polarizations of gravitons.

dure. Needless to say, these equations have much greater degree of non-linearity, and this diminishes our chances of constructing an exact solution. In particular, there are no known vacuum (i.e. with the zero right hand side in the generalized Einstein equations) BH-type solutions. However, this particular complication can be tackled by assuming that at least at large distances, known BH configurations can be taken as an approximation for the background. Surely, this still results in way more complicated perturbations compared to a local theory of GR.

The following setup forms the content of this paper: we resort to the case of a static BH solution and approximate it by a Schwarzschild metric at large distances. This allows us to substitute the vanishing Ricci tensor and the Ricci scalar for the background quantities. We expect that perturbations in such a background in a modified, infinite derivative gravity theories will contain not only standard real mass fields but also novel fields with complex masses squared mentioned in the previous paragraph. Such fields are expected to produce previously non-existing effects, and the study of the corresponding systems comprises the main part of the current paper. A static BH configuration is sufficient to highlight the effects of gravity modification on QNMs of $\mu$BHs.

A case of rotating BHs, charged BHs, or other physically important configurations are not discussed in details in the present paper. Effects of higher and infinite derivative gravity models can be captured already by a static Schwarzschild BH configuration. Nevertheless fulfilling this gap will be an important question for future projects. Recently several studies have appeared addressing similar questions in the fourth order gravity [101, 102].

The paper is organized as follows. In Section 2 we describe emergence of BISs in a scalar toy model and in AID gravity; in Section 3 we discuss properties of QNMs of scalar fields with complex and radial dependent masses; in Section 4 we collect our numeric results obtained mainly using a continued fraction technique outlined in the Appendix. We then recollect and summarize our analysis in the Discussion and Outlook Section highlighting directions for future study.

## II.  NON-PERTURBATIVE GRAVITY, BACKGROUND INDUCED STATES (BIS), AND COMPLEX-VALUED MASSES.

Both in AID gravity and asymptotically safe gravity the relevant part of the action on which these proposals are based has the following form:

$$S = \int d^D x \sqrt{-g} \left( \frac{M_P^2}{2} R + \frac{\lambda}{2} \left( R\mathcal{F}(\Box)R + R_{\mu\nu}\mathcal{F}_2(\Box)R^{\mu\nu} + R_{\mu\alpha\nu\beta}\mathcal{F}_4(\Box)R^{\mu\alpha\nu\beta} \right) + \dots \right) \quad (1)$$

where $M_P$ is the Planck mass and $\lambda$ is a constant useful to track the GR limit. We use throughout this paper a metric with the signature $(-, +, +, +)$, work in 4 space-time dimensions, and mainly use natural units $\hbar = c = 1$. $\mathcal{F}_i(\Box)$ are higher or infinite derivative operators (also often named form-factors) which can be represented as Taylor series at zero. $\Box = \nabla^\mu \nabla_\mu$ is a covariant d'Alembertian and $\nabla_\mu$ is a covariant derivative. An analyticity at zero allows having usual EFT of GR as a local limit. Moreover, due to power-counting arguments there should be a scale entering as $\Box/\mathcal{M}^2$ which we name the non-locality scale. In what follows we set it to be unity before any quantitative analysis. Finally, dots in the above action indicate terms which do not contribute to the propagator.

Equations of motion are rather complicated [103] but one can easily see that Minkowski metric is a solution. Computing a propagator around the Minkowski space-time one can find for the spin-2 graviton part [79]

$$\Pi_{spin-2} \sim \frac{1}{\Box \left( 1 + \frac{\lambda}{M_P^2} \Box(\mathcal{F}_2(\Box) + 2\mathcal{F}_4(\Box)) \right)} \quad (2)$$

The latter form which suggests that for the case of an infinite number of derivatives one can demand

$$1 + \frac{\lambda}{M_P^2} \Box(\mathcal{F}_2(\Box) + 2\mathcal{F}_4(\Box)) = e^{\omega(\Box)}$$

with $\omega(\Box)$ some entire function and this will give no new finite roots. Otherwise, new poles resulting in Ostrogradski ghosts [104] would appear. Algebraically, an exponent of an entire function of derivative operator in the propagator is the only way to circumvent the ghosts problem. A not so obvious and puzzling observation is that given some function $\omega(\Box)$ is fixed to have the Minkowski background well behaving with just a single graviton excitation in the spectrum. Expanding around any other background one would break this nice exponential of an entire function construction and generate new states – the Background Induced States (BISs).

One can track how and which new excitations can appear around a different background in a gravity theory with higher derivatives by resorting to a scalar field example. One can consider an action

$$S = \int d^4x \left( \frac{1}{2}\varphi(\Box - m^2)f(\Box)\varphi - V(\varphi) \right)$$

Then one obviously sees that the above action in the case $f(\Box) = 1$ represents a standard scalar field Lagrangian describing a single scalar field excitation of mass $m$. The new ingredient $f(\Box)$ is the higher derivative modification. We take it analytic at zero function such that $f(0) = 1$ to preserve the normalization. If the Taylor series at zero has a finite number of terms, then one inevitably will encounter the Ostrogradski ghosts and an infinite derivatives can help evading them when $f(\Box) = \exp(2\sigma(\Box))$ where $\sigma(\Box)$ is an entire function. Then a propagator does not get any new pole for finite values of momenta as an inverse of an exponent is again an exponent. This way, however, we create an essential singularity in the propagator at infinity which requires a special treatment from the point of view of QFT computations for scattering amplitudes [74, 75].

The just presented nice construction gets broken in a nontrivial vacuum. Indeed, given a vacuum $\varphi = \varphi_0 \neq 0$ one can expand the above action around it to yield:

$$S_2 = \int d^4x \left( \frac{1}{2}\delta\varphi(\Box - m^2)f(\Box)\delta\varphi - \frac{1}{2}V''(\varphi_0)\delta\varphi^2 \right),$$

and the corresponding propagator for excitations now is

$$\Pi_2 = \left[ (\Box - m^2)f(\Box) - V''(\varphi_0) \right]^{-1}.$$

Unless $V''(\phi_0) = 0$ we effectively have created infinitely many excitations as the above expression can be represented as an infinite product using the Weierstrass decomposition for entire functions. Given that in the original action, all the Taylor series coefficients of $f(\Box)$ were real, as they are some physical parameters of our model, corresponding zeros of the above propagator are either real or complex conjugate. New real poles would correspond to ghosts while complex poles are still more curious. Even though this mess can be avoided in a special case $V''(\varphi_0) = 0$ it is impossible in general and especially absolutely not the case if $\varphi_0$ is not a constant but rather a non-trivial variable background. It is important to mention that such new modes would always be problematic around Minkowski background as they always will have growing modes but can be tamed around non-trivial background and made such that no growing modes appear (see [71] for an example of the de Sitter space-time).

The same story happens in a gravity action with form-factors $\mathcal{F}_i(\Box)$ adjusted such that no new poles appear around a Minkowski space-time upon expanding the action to the second order around any other background. To see how this is happening one can consider an action

$$S = \int d^4 x \sqrt{-g} \left[ \frac{M_P^2}{2} R + \frac{\lambda}{2} (R \Box R + R_{\mu\nu} \Box R^{\mu\nu})) \right] \tag{3}$$

This model example obviously contains ghosts but this is not the thing which should bother us. We only want to see that upon expanding to a second order around some background we will generate terms which were not present around Minkowski space-time. Choosing a Schwarzschild metric as a test background (which moreover is a solution because locally outside of a BH it has $R_{\mu\nu} = R = 0$) we get for the trace of the metric tensor perturbations $h$

$$\delta R_{\mu\nu} \Box \delta R^{\mu\nu} \sim = h(3\Box^3 + D^\mu R^\sigma_{\mu\alpha\nu} D^\alpha D_\sigma \partial^\nu + D^\mu D^\alpha R^\sigma_{\mu\alpha\nu} D_\sigma \partial^\nu)h \tag{4}$$

In Minkowski space-time this would be simply $\sim h\Box^3 h$ therefore supporting our claim that the propagator structure is going to change. This will dismantle a nice exponent of an entire function construction and lead to new excitations with some of them having complex masses[2]. Furthermore, the masses which are generated will be coordinate-dependent quantities simply because the curvature tensor components are in general coordinate dependent. Strictly speaking, in order to decompose the quadratic action into separate actions for BISs one needs to assume that the background is smooth enough, such that WKB approximation is valid. It is not clear how to deal with the case when WKB approximation is not working. It is easily seen above as the Riemann tensor components even in a simple example of the Schwarzschild BH depend on the radius $r$.

As mentioned in the Introduction, a method of analyzing a behavior of test fields in a BH background can be used to capture main properties of perturbations of a BH. We therefore are set to explore new effects and possible signatures of BH perturbations due to a presence of complex and moreover $r$-dependent mass states in the spectrum. Our focus will be centrally-symmetric static backgrounds.

---

[2] To see that masses indeed can easily become complex one can also think about a very simple scalar field action with six derivatives $S = \int d^4 x \left( \frac{1}{2} \varphi \Box (\Box - m^2)(\Box - 2m^2)\varphi - V(\varphi) \right)$. It describes three excitations with masses squared $0$, $m^2$, $2m^2$, but given there is non-trivial vacuum such that $V''(\varphi) = m^6$ one would get a pair of complex conjugate masses.

## III.   PROPERTIES OF QNMS GENERATED BY A COMPLEX MASS SCALAR FIELD

In general, one should start by solving a free field equation in a BH background, i.e. the Klein-Gordon equation. Even though relevant equations were obtained in numerous papers (see [8] for a review) it is instructive to reiterate it briefly for the sake of used terminology and notations. For a scalar field $\Phi$ with mass $\mu$ in a background with the metric $g_{\mu\nu}$ its covariant Klein-Gordon equation is

$$(\nabla^\nu\nabla_\nu - \mu^2)\Phi(t,r,\theta,\phi) = 0 \tag{5}$$

where $\nabla_\nu$ is the covariant derivative. We aim to study centrally symmetric and static backgrounds. We resort to Schwarzschild coordinates which are time $t$, radius $r$, and angles $\theta, \phi$. In what follows, we work in the approximation of the Schwarzschild background with the line element

$$ds^2 = -f(r)dt^2 + f(r)^{-1}dr^2 + r^2 d\Omega_2^2, \quad f(r) = 1 - \frac{2GM}{r}. \tag{6}$$

This approximation is valid as long as we assume that gravity is significantly modified only on length scales much less than the Schwarzschild radius. Upon a standard separation of variables, we have

$$\Phi(t,r,\theta,\phi) = \sum_{l,m} \Psi(t,r)\frac{Y_{lm}(\theta,\phi)}{r} \tag{7}$$

where $Y_{lm}(\theta,\phi)$ are spherical harmonics and $l$ and $m$ stand for the angular and azimuthal number. Using this variable separation and introducing in passing the tortoise coordinate $r_*$ given by

$$dr_* = \frac{dr}{1 - \frac{2GM}{r}} \tag{8}$$

we come to the following equation

$$\left[\frac{\partial^2}{\partial t^2} - \frac{\partial^2}{\partial r_*^2} + V(r)\right]\Psi(t,r) = 0 \tag{9}$$

with

$$V(r) = \left(1 - \frac{2GM}{r}\right)\left(\frac{l(l+1)}{r^2} + \frac{2GM}{r^3} + \mu^2\right). \tag{10}$$

QNMs are solutions of the latter wave equation (9), with time dependence $\Psi(t, r_*) \sim e^{-i\omega t}\Psi(r_*)$ satisfying boundary conditions of a pure in-going wave at the horizon and pure out-going wave at spatial infinity. That is[3]

$$\Psi(t, r_*) \sim \begin{cases} e^{-i\omega t - i\omega r_*}, & r_* \to -\infty \\ e^{-i\omega t + i\sqrt{\omega^2 - \mu^2}r_*}, & r_* \to +\infty \end{cases} \tag{11}$$

The remaining equation becomes

$$\frac{\partial^2}{\partial r_*^2}\Psi(r_*) + (\omega^2 - V(r))\Psi(r_*) = 0 \tag{12}$$

The novel aspect we are going to analyze is a situation when $\mu$ is a complex valued parameter and also when mass depends on the radius, i.e. $\mu = \mu(r)$, in a centrally symmetric background. There are two comments in order here. First, direct and systematic derivation of the corresponding masses from the bare action of AID or from quantum effective action in Asymptotically safe gravity is a very difficult task. It can be performed in a scalar field non-local toy-model. It was done in a de Sitter background [71] and seems to be an outstanding problem in a more complicated situation. What is clear though is that values of masses depend on two factors: background metric and a shape of the form-factors. The latter means that a characteristic scale in determining these masses would be the non-locality scale $\mathcal{M}$. While its value is not known, a study of the primordial inflation in this framework [72, 73] reveals that $\mathcal{M}$ should be much greater then an inflation scale. Second, an explicit radial dependence of the resulting masses is also difficult to compute but since they are formed as Riemann tensor coefficients and its powers we can suggest that a plausible dependence is $\sim 1/r^\gamma$ for some positive and most likely integer $\gamma$.

## A. Possible growing modes

It is obvious from (11) that for $\mathrm{Im}(\omega) > 0$, the perturbation will grow with respect to time. However, such modes in a standard case can be excluded by the argument first presented in [105] which we briefly review here. The time-dependent equation is

$$\frac{\partial^2 \Psi}{\partial t^2} - \frac{\partial^2 \Psi}{\partial r_*^2} + V(r)\Psi = 0 \tag{13}$$

---

[3] Note that the coordinate $r_*$ varies from $(-\infty, +\infty)$ as $r$ changes from the Schwarzschild radius to infinity.

If we impose the QNM boundary condition and take $\text{Im}(\omega) > 0$ then $\Psi$ will decay exponentially with respect to $r$. Multiplying the latter equation by $\frac{\partial \Psi^*}{\partial t}$ and doing an integration by parts over $r$ (the boundary term can be thrown away since $\Psi$ decays exponentially in both spatial infinities), we obtain

$$\int_{-\infty}^{+\infty} \left( \frac{\partial \Psi^*}{\partial t} \frac{\partial^2 \Psi}{\partial t^2} + \frac{\partial \Psi}{\partial r_*} \frac{\partial^2 \Psi^*}{\partial t \partial r_*} + V(r)\Psi \frac{\partial \Psi^*}{\partial t} \right) dr_* = 0 \tag{14}$$

Taking a sum of the above expression and its complex conjugate, we get

$$\frac{\partial}{\partial t} \int_{-\infty}^{+\infty} \left( \left| \frac{\partial \Psi}{\partial t} \right|^2 + \left| \frac{\partial \Psi}{\partial r_*} \right|^2 + V(r)|\Psi|^2 \right) dr_* = 0 \tag{15}$$

Notice that this formula is correct only if $V(r)$ is real. In that case the integral itself is a constant with respect to time. Then, since for any bounded static solution to (13) the corresponding energy integral is always convergent, and accounting that QNMs do exactly correspond to bounded solutions, one can conclude that growing in time solutions are excluded.

However, once we introduce complex parameters in the potential, the above arguments will fail because adding to (14) its complex conjugate will no longer give a total derivative with respect to time. So, in the case of complex mass parameters, unstable growing in time modes can not be excluded anymore, and we must put constraints to avoid them in order to ensure the stability of a BH background.

### B. Late-time tails for complex masses fields and causality issue

One of the method analyzing solutions to the wave equation (9) and in particular QNMs is to consider Green's function which poles in fact represent QNMs [7]. This allows to find a behavior of the late-time tails in a BH background, which decreases as $t^{-\alpha}$ for some positive $\alpha$ and thus dominate the late-time behavior of perturbations. Such contribution comes from a branch cut due to singularities of the Green's functions [24]. In terms of Green's function, the solution to wave-like equation (9) is given by

$$\Psi(r_*, t) = \int [G(r_*, r_*', t)\Psi_t(r_*', 0) + G_t(r_*, r_*', t)\Psi(r_*', 0)]dr_*' \tag{16}$$

where the retarded Green's function $G$ satisfies

$$\left[ \frac{\partial^2}{\partial t^2} - \frac{\partial^2}{\partial r_*^2} + V(r) \right] G(r_*, r_*', t) = \delta(t)\delta(r_* - r_*') \tag{17}$$

A time Fourier transform $\tilde{G}(r_*, r'_*, \omega)$ can be written as

$$\tilde{G}(r_*, r'_*, \omega) = \frac{1}{\tilde{\Psi}_1 \partial_{r_*} \tilde{\Psi}_2 - \tilde{\Psi}_2 \partial_{r_*} \tilde{\Psi}_1} \begin{cases} \tilde{\Psi}_1(r'_*, \omega)\tilde{\Psi}_2(r_*, \omega), & r_* > r'_* \\ \tilde{\Psi}_1(r_*, \omega)\tilde{\Psi}_2(r'_*, \omega), & r_* < r'_* \end{cases} \tag{18}$$

where $\tilde{\Psi}_1$ is the Fourier transform of the solution satisfying the in-going wave condition at the event horizon, and $\tilde{\Psi}_2$ is the Fourier transform of the solution satisfying the out-going wave at infinity. Notice that the boundary condition at spatial infinity implies that $\tilde{\Psi}_2 \to e^{\sqrt{\omega^2 - \mu^2} r_*}$ so there are two singularities on $\tilde{\Psi}_2$, located at $\pm a$ given $\mu^2 = a^2$ (we introduce this new notation for a consistency with computations hereafter). For physical considerations we choose $\tilde{\Psi}_1$ to be analytic so that it doesn't contribute to the late-time tail [24].

The next step is to determine a contour for the inverse Fourier transform

$$G(r_*, r'_*, t) = \frac{1}{2\pi} \int_{-\infty}^{\infty} \tilde{G}(r_*, r'_*, \omega)e^{-i\omega t}d\omega \tag{19}$$

In the case of a complex $\mu$ depicted in FIG. 1, two singularities at $\mu = \pm a$ are located above and below the real axis, corresponding to a branch cut passing through the origin (the red line). In a language of Green's functions $G(r_*, r'_*, t)$, causality requires that the contribution from $t < 0$ must vanish, as the perturbation is performed at $t = 0$ and cannot propagate backward in time. Therefore, we expect $G(r_*, r'_*, t) = 0$ for $t < 0$. Then, when $t < 0$, the integration contour of (19) must be closed on the upper half-plane to ensure that the integral converges to zero along the large arc of a sufficiently large radius. However, the presence of a complex mass $\mu$ implies that at least one of $\mu = \pm a$ will be located in the upper half-plane. Consequently, this contour will necessarily include one singularity, leading to $G(r_*, r'_*, t) \neq 0$ for $t < 0$. Therefore, for a constant complex mass $\mu$, we always will observe a causality violation.

This consideration provides a very strong hint that if complex masses are present, they at least cannot be constants but rather be variable radius-dependent parameters. Indeed, since the branch cut for the late-time tales essentially comes from the boundary conditions at spatial infinity, a natural idea to resolve the causality issue but still keeping complex masses in the game is to require that the imaginary part of a mass vanishes at spatial infinity[4]

---

[4] The same requirement applied to the case of de Sitter space implied vanishing all imaginary parts of BISs in the limit of infinite de Sitter radius or in the limit of flat spacetime.

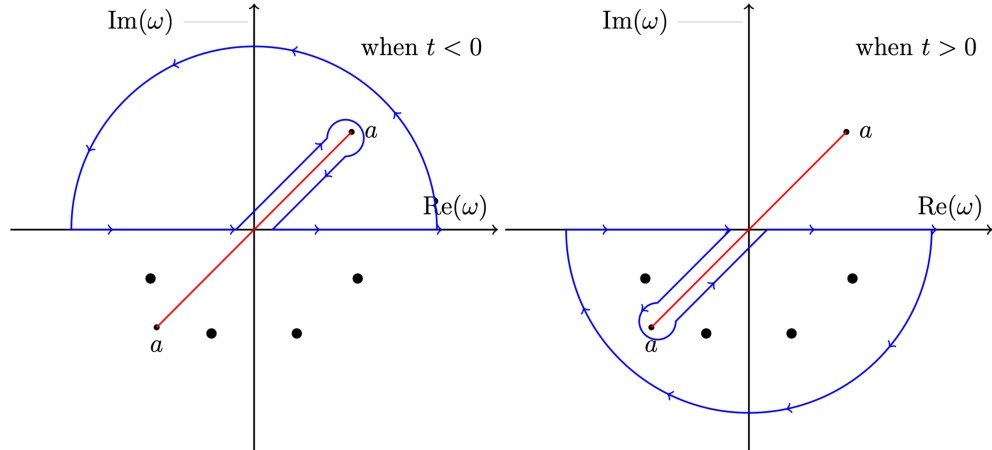

FIG. 1: Integration contour of the inverse Fourier transform (19) in the case of a complex

mass $\mu^2 = a^2$ where black dots denotes QNMs and the red line denotes the branch cut

Recall that having complex mass states on top of flat spacetime is not consistent even with classical stability. For this reason, it is not surprising that there is a problem with constant complex-valued masses because BH background is asymptotically flat at spatial infinity. But one can expect that this problem can be resolved for imaginary parts vanishing far away from the BH. In this case, the corresponding boundary condition at spatial infinity becomes

$$\tilde{\Psi}_2 \to e^{i\sqrt{\omega^2 - \mu(r)^2} r_*} \to e^{i\sqrt{\omega^2 - \text{Re}(\mu^2)} r_*}, \ r_* \to +\infty \tag{20}$$

which restores a real mass situation.

An example of the analysis aimed at verification of causality can be performed analytically at least for large $r$ if we take

$$\mu^2 = a^2 + \frac{ic}{r^2} \tag{21}$$

where $a$ and $c$ are some real parameters and $c$ is dimensionless. Since the causality is broken due to the boundary conditions at spatial infinity, we only need to study the behavior of the solution in the faraway asymptotic region. We then neglect terms of order $O(1/r^3)$ for simplicity and equation (9) in $r$ coordinate becomes a wave-like equation as follows

$$\frac{\partial^2}{\partial r^2}\Psi - \left(\left(a^2 + \frac{ic}{r^2} - \omega^2\right) - \frac{1}{r}(-a^2 + 2\omega^2) - \frac{2\omega^2 - l(l+1)}{r^2}\right)\Psi = 0 \tag{22}$$

where we normalize the Schwarzschild radius to a unit, i.e. $2GM = 1$ for simplicity. Solutions

to this equation are

$$A_1 = \tilde{M}\left(\frac{2\omega^2 - a^2}{2\sqrt{a^2 - \omega^2}}, \frac{1}{2}i\sqrt{-4l^2 - 4l + 8\omega^2 - 4ic - 1}, 2r\sqrt{a^2 - \omega^2}\right)$$

$$A_2 = \tilde{W}\left(\frac{2\omega^2 - a^2}{2\sqrt{a^2 - \omega^2}}, \frac{1}{2}i\sqrt{-4l^2 - 4l + 8\omega^2 - 4ic - 1}, 2r\sqrt{a^2 - \omega^2}\right) \tag{23}$$

where $\tilde{M}(k, u, z)$ and $\tilde{W}(k, u, z)$ are the Whittaker functions and their asymptotic behavior at infinity is as follows

$$\tilde{W}(k, u, z) \rightarrow e^{-\frac{z}{2}} z^u$$

$$\tilde{M}(k, u, z) \rightarrow e^{\frac{z}{2}} z^{-u} \frac{\Gamma(1+2k)}{\Gamma(1/2+k-u)} + e^{-\frac{z}{2}} z^u (-1)^{\frac{3}{2}-k+u} \frac{\Gamma(1+2k)}{\Gamma(1/2+k+u)}$$

$\Psi_2$ is a linear combination

$$\Psi_2 = a_1(\omega) A_1 + a_2(\omega) A_2 \tag{24}$$

and the boundary condition for QNMs implies that $a_1(\omega)$ and $a_2(\omega)$ should be chosen such that $\Psi$ has no singularity on the upper half-plane at spatial infinity. Furthermore, since there is no $r$ dependence of the singularities of $A_{1,2}$, there will be no branch cut on the upper half plane for any $r$ and thus causality is preserved by choosing contour as in FIG. 2.

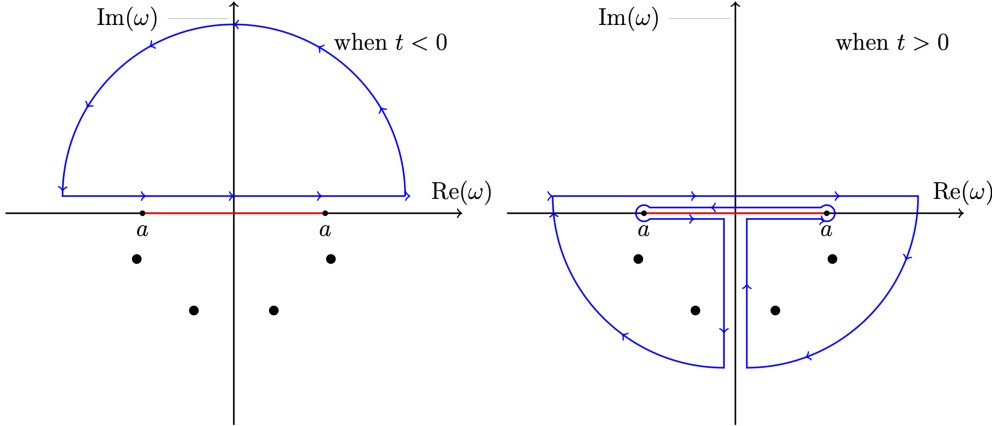

FIG. 2: Integration contour of the inverse Fourier transform (19) in the asymptotic regime $r \rightarrow \infty$ when $\mu^2 \rightarrow a^2$ and $a$ is real, and where black dots denotes QNMs and the red line denotes the branch cut

## IV. NUMERICAL RESULTS FOR GENERAL RADIUS DEPENDENT COMPLEX MASSES

A more general case of a curvature-dependent complex mass cannot be easily treated analytically and it is natural to resort to a numeric analysis. Now we introduce a more general example of a variable mass

$$\mu(r)^2 = a^2 + \frac{b}{2GMr} + \frac{c}{r^2} + \frac{2GMd}{r^3} \tag{25}$$

where $a$ is real constant part of the mass while $b, c, d$ are some parameters which can contain real and imaginary parts and they all three are dimensionless for simplicity. Even though we introduce an ad hoc potential it will allow us to see viability of our computations for different dependence of a scalar field mass on the radius. The potential becomes as follows

$$V(r) = \left(1 - \frac{2GM}{r}\right)\left(\frac{2GM}{r^3} + \frac{l(l+1)}{r^2} + a^2 + \frac{b}{2GMr} + \frac{c}{r^2} + \frac{2GMd}{r^3}\right) \tag{26}$$

In what follows we are going to compute the QNM spectrum of equation (12) with boundary conditions (11) for the above written potential numerically using the continued fraction method. Details of this method are collected in Appendix A. We normalize $2GM = 1$ in our calculations. Also as mentioned above we mainly target in our consideration $\mu$BHs with a mass of order of a Planck mass and such a normalization is not hiding real physical quantities.

QNMs are typically ordered by the magnitude of the negative imaginary part of their frequencies $\omega$ for the same angular number $l$, with smaller values corresponding to lower overtones. We refer to the mode with the smallest negative imaginary part as the fundamental (or first) mode, while subsequent modes starting from the second one are labeled as the $n$-th overtone.

FIG. 3 illustrates our numerical results for the first three overtones with $l = 0$ and with no real part of the mass, i.e. $a = 0$, while parameters $b$, $c$, and $d$ have purely imaginary varying values. First of all, we see that frequencies got displaced from there standard values denoted as red dots. Moreover, we see that when the imaginary part of $b$, $c$, $d$ exceeds some critical value $b_m \sim 1$, $c_m \sim 0.75$, $d_m \sim 1$, the imaginary part of the frequency of the first overtone becomes positive, which means that instable configurations can indeed exists in this model.

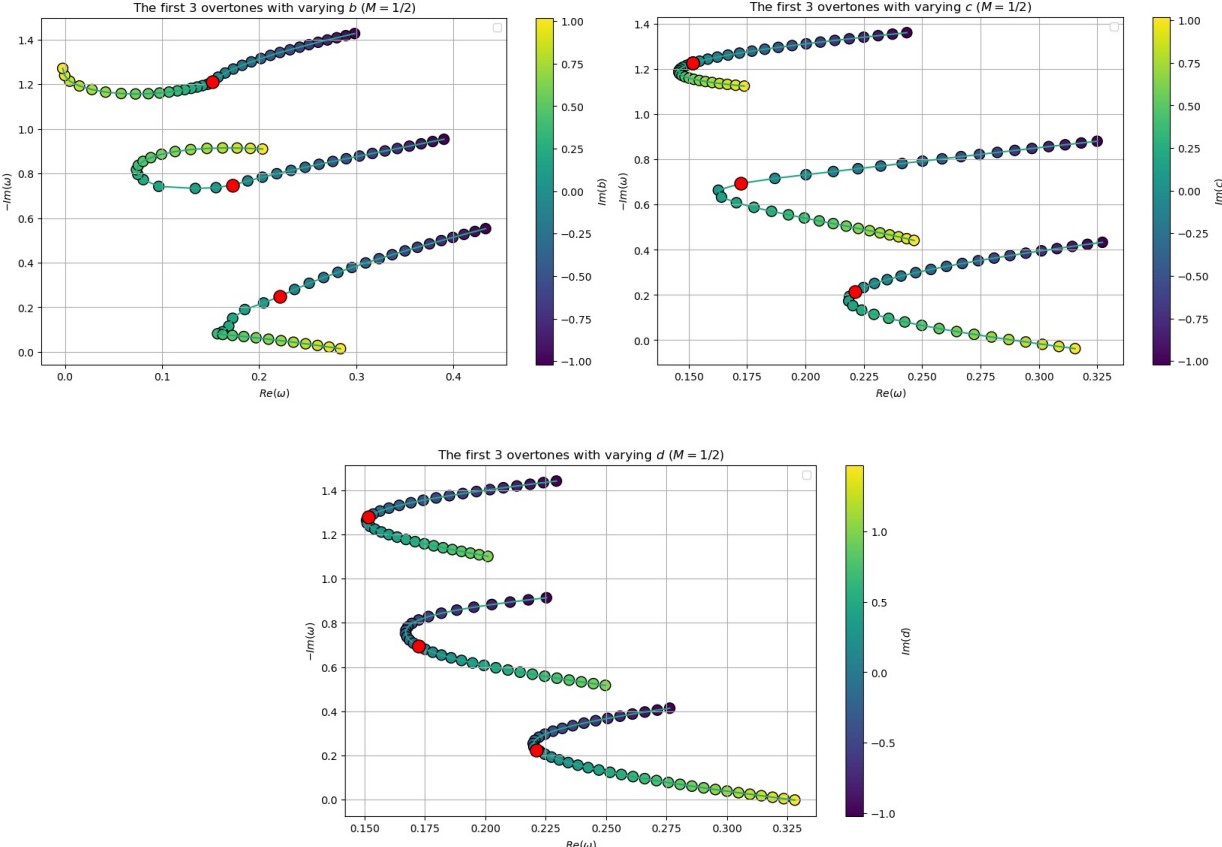

FIG. 3: QNM for $a = 0$ and $l = 0$ with varying pure imaginary parameters $b$ (top left), $c$ (top right), and $d$ (bottom) where the color of points represents the value of the corresponding varying parameter as shown in the bar to the right of each plot

We then compute the fundamental mode for several values of $a$ and vary pure imaginary parameters $c$ (left), $d$ (right) in FIG. 4. In each plot, there are three curves with red dots corresponding to $b = c = d = 0$. In this Figure a cyan curve corresponds to $a = 0$, a purple curve to $a = 0.15$ and a blue curve to $a = 0.3$. In comparison with a standard case represented by red dots, one immediately notices that the presence of complex mass parameters introduces many more possible values in the QNM spectrum.

Further numerical results show that for example for a fixed $c$ we can increase $a$ to pull an unstable mode back to the stable region. Fig. 5 shows how the first overtone changes when $c = 2$ and $a$ changes from 0 to 0.95. For $a = 0$ this mode is unstable, but as we increase $a$, the imaginary part gradually moves towards a stable region. At $a = 0.95$, it finally comes back to the upper half plane and becomes stable. Similar behavior applies to all unstable modes.

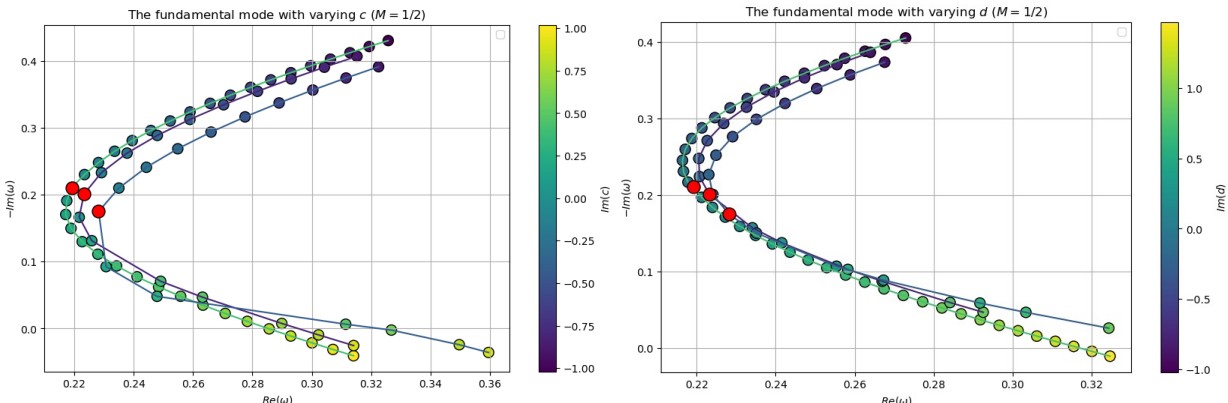

FIG. 4: Fundamental mode for varying $c$ (left) and $d$ (right), with $a = 0,\ 0.15,\ 0.3$

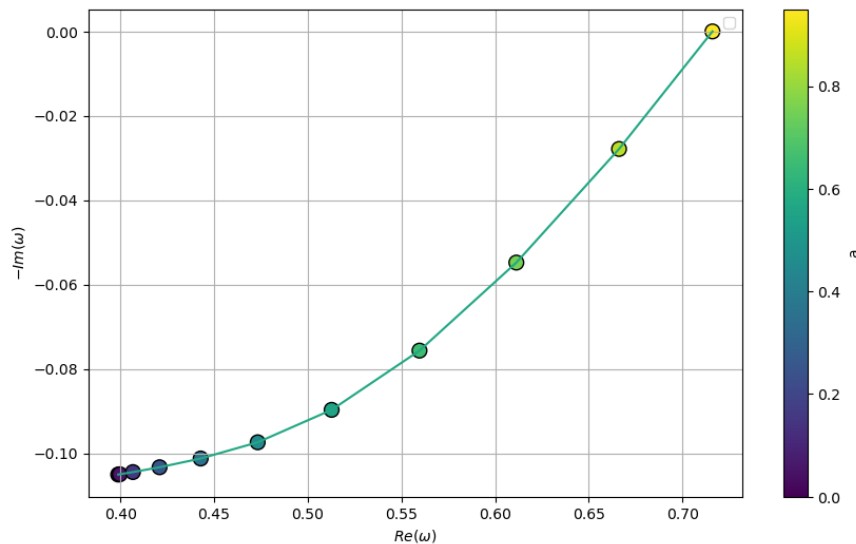

FIG. 5: The first overtone with $c = 2$ and increasing $a$

To see a presence of stable and unstable regions depending on the values of the complex parameters and $a$, we perform more computations with different $c$ and $d$. With each $c$, $d$, we extract the limiting value of $a$ such that the first overtone reaches the stable region. These results are shown in FIG. 6. The left plot is for varying pure imaginary $c$ (with $b = d = 0$) and the right one is for varying pure imaginary $d$ (with $b = c = 0$). In this Figure we specifically take values of $c$ and $d$ greater than 1 as smaller parameters produce stable modes anyway according to results explained above.

As a short warp up of this Section, we have shown that for small values of imaginary parts of complex parameters of this model can have both stable and causal QNMs. This essentially prompts for constraints on parameters $a, b, c, d$ which are determined by the model. A more

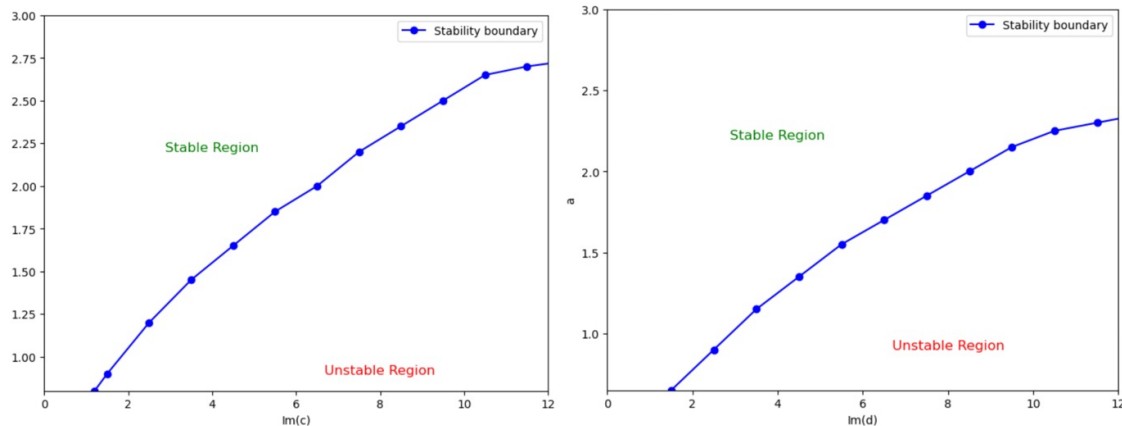

FIG. 6: Stability region of $a$ and $c$ (left), $d$ (right)

detailed discussion follows in the next section.

## V.   DISCUSSION AND OUTLOOK

QNMs of $\mu$BHs in non-perturbative quantum gravity candidates has been discussed in the present manuscript with the most attention paid to the effective Background Induced States (BISs). The emergence of these states can be traced back to the structure of the infinite derivative operators required to formulate a UV complete and unitary gravity theory. These states in general are created as pairs of coherent (featuring the mutually complex conjugate pairs of creation-annihilation operators) fields with complex conjugate complex masses. The presence of such states does not have to be always harmful for unitarity, as it was discussed in [74, 75]. These states can be such that no growing modes exist. However, they do create new effects in different considerations, for example in inflation [72, 73] and here we have examined their influence on the spectrum of QNMs. The nature and origin of BISs were explored in Section 2 while their essential properties relevant for the present paper were explored in Section 3.

In this work, we have taken the path of considering QNMs of BHs due to the perturbations of external fields on top of the BH background. Namely, we examine the effects emerging from perturbations of the scalar fields with complex-valued curvature-dependent masses. In the described setup, these effects are relevant for micro-BHs ($\mu$BHs) while for astrophysical BHs possible changes in the QNMs spectra are very much suppressed [49]. In this paper, we are concentrating on the $\mu$BHs with masses of order of a Planck mass. For simplicity,

we choose a static nearly Schwarzschild BH background, in order to study how the presence of BISs affects BH quasinormal modes. A nearly Schwarzschild approximation should be valid outside the horizon as long as gravity modifications touch only extreme UV regions. A static BH is already capable to demonstrate effects of gravity modification which is the main focus of the present paper.

We have established in Section 3 that the corresponding complex masses of scalar fields must have at least their imaginary part to be radial-dependent. If this is not the case, the causality is broken according to the late-tails analysis performed in Section 3. We then switched to a numerical analysis in Section 4 due to the complexity and impossibility of deriving analytic expression. Computations were done mainly using the continued fraction method [106]. We have observed that complex masses lead to modified QNMs frequencies compared to a GR setup. Here we observe at least a similarity with the complex momenta modes discussed in [107]. Advance in understanding of such a relation can be an interesting question for future study.

In addition, we particularly addressed the question of the stability of possible QNMs. Indeed, we have found that unstable growing in time QNMs can exist as long as fields with complex masses are present, contrary to the argument outlined in [105] which forbids such growing modes in GR. We saw that unstable states can be avoided by means of choosing appropriate parameters for the complex masses values. These parameters, in turn, appear from the original model (1), and are mainly determined by the shapes of form-factors $\mathcal{F}_X(\Box)$. Tracing explicitly and rigorously a relation between the form-factors and masses of QNMs is a very complicate task. We certainly hope to address it in the near future but definitely its implementation will require a significant time and effort. However, we are enthusiastic to say that this could be one more constraint on the non-perturbative quantum gravity candidates.

Interestingly, we see that depending on what is the primary input, parameters of the model or masses of $\mu$BHs different conclusions can be made. Indeed, given that we have certainty and assurance in deriving parameters governing BISs we can constrain the lower mass limit of a $\mu$BH. Let's presume that imaginary parts of mass parameters at the Schwarzschild radius where they are the most important are of order of the Planck mass or more. Then, analyzing the stability of QNMs with complex radial dependent masses we observe that larger imaginary part of parameters in the radial dependent part of mass should be correlated with the real constant mass parameter to acquire a stable configuration. Looking at FIG. 6

we see this dependence in particular cases of radial dependence. Resorting to the situation when the imaginary part decays inverse proportional to the radius squared (parameter $c$ in our analysis) and assuming very roughly that a stable/unstable region separation is linearly proportional following the numerical results on the left plot in FIG. 6 we can do a quick dimensional estimation of the $\mu$BH lower mass bound.

Accounting that the Schwarzschild radius is growing linearly with respect to the BH mass, and assuming that the real part of a field mass (the one we would observe at spatial infinity) cannot exceed a Planck mass we can easily deduce that the BH mass should not be lower than a Planck mass. This straightforwardly follows from the shape of the potential (26) and the left plot in FIG. 6. However, on the other hand we can go to the opposite direction and claim that $\mu$BH masses are given external parameters, this will put constraints on the BISs masses parameters and in turn will constrain parameters and form-factors of the full gravity theory.

As an important future direction we will study other spherically-symmetric solutions and especially rotating BHs and corresponding QNMs in the framework of UV complete gravity theory approaches. Being important these configurations are not expected to add much to the effects already observed using the Schwarzschild BH example. Namely, we expect that the only healthy possibility is radius or in general coordinate dependent masses of BIS-s, and also we anticipate that QNM-s can become unstable requiring additional restrictions on model parameters. It is also important and necessary to study gravitational perturbations of BH configurations in gravity modifications motivated by the Quantum Gravity constructions.

**ACKNOWLEDGMENTS**

AK and AT would like to thank organizers of the "Quantum Gravity 2024" program in NORDITA where part of this work has been done and especially Steve Giddings for valuable discussions on the topic. AT is grateful to Ivano Basile, Gia Dvali, and Arkady Tseytlin for illuminating conversations on the problem of background-induced states. The work of AT was supported by the National Natural Science Foundation of China (NSFC) under Grant No. 12347103.

## Appendix A: Continued fraction method

Here we outline the Leaver's continued fractions method [18, 106] tailored to our particular equation and potential in order to to compute the corresponding QNMs. The strategy is to construct a function that satisfies both boundary conditions and fits the behavior in the middle region by introducing a Frobenius series which doesn't affect the boundary behavior, thus getting a solution satisfying both boundary conditions in terms of a series. QNM condition essentially becomes a condition to make this series convergent.

First, since we assume that a variable mass vanishes at spatial infinity, the boundary conditions are equivalent to that of a field with a constant real mass $\mu_\infty$, which is

$$\Psi(r) \sim C_-(r - 2GM)^{-2iGM\omega}, \quad (r \to 2GM)$$
$$\Psi(r) \sim C_+ e^{i\chi r} r^{i\frac{GM\mu_\infty^2}{\chi}}, \quad (r \to \infty) \tag{A1}$$

where $\chi = \lim_{r \to \infty} \sqrt{\omega^2 - \mu(r)^2} = \sqrt{\omega^2 - \mu_\infty^2}$. Then $\Psi(r)$ can be written as

$$\Psi(r) = e^{i\chi r} r^{2iGM\chi + iGM\mu_\infty^2/\chi}(1 - \frac{2GM}{r})^{-2iGM\omega} \sum_n a_n (1 - \frac{2GM}{r})^n \tag{A2}$$

where we have introduced an extra factor $r^{2iM\omega}$ to make the term $(r - 2M)^{-2iM\omega}$ tend to a unit at spatial infinity so that it will not influence a behavior at spatial infinity. Substituting this to equation (12) written in terms of $r$ with potential (26) gives us a four-term recursion relation

$$\alpha_0 a_1 + \beta_0 a_0 = 0$$
$$\alpha_1 a_2 + \beta_1 a_1 + \gamma_1 a_0 = 0 \tag{A3}$$
$$\alpha_n a_{n+1} + \beta_n a_n + \gamma_n a_{n-1} + \delta_n a_{n-2} = 0, \quad n > 1$$

with

$\alpha_n = (n+1)\chi^2(-4iGM\omega + n + 1)$

$\beta_n = \chi(-\chi(b + c + d - 12G^2M^2\omega^2 - 12iGMn\omega - 4iGM\omega + l^2 + l + 3n^2 + 2n + 1)$
$\quad + 4G^2M^2\chi^3 + 3GM\chi^2(4GM\omega + 2in + i) + GM\omega^2(4GM\omega + 2in + i))$

$\gamma_n = \chi^2 \left(c + 2d - 18G^2M^2\omega^2 - 12iGM(n-1)\omega - 8iGM\omega + l^2 + l + 3(n-1)^2 + 4(n-1) + 2\right)$
$\quad - 5G^2M^2\chi^4 - G^2M^2\omega^4 - GM\chi^3(16GM\omega + 8i(n-1) + 5i) - GM\chi\omega^2(8GM\omega + 4i(n-1) + 3i) \tag{A4}$

$\delta_n = -\chi^2 \left(d - 6G^2M^2\omega^2 - 4iGM(n-2)\omega - 4iGM\omega + (n-2)^2 + 2(n-2) + 1\right) + G^2M^2\chi^4$
$\quad + G^2M^2\omega^4 + 2GM\chi^3(2GM\omega + i(n-2) + i) + 2GM\chi\omega^2(2GM\omega + i(n-2) + i)$

where $\chi = \sqrt{\omega^2 - \mu_\infty^2}$.

Further, we reduce the previous recursion to a three-term recursion relation

$$\alpha_0' a_1 + \beta_0' a_0 = 0$$

$$\alpha_n' a_{n+1} + \beta_n' a_n + \gamma_n' a_{n-1} = 0, \ n > 0 \tag{A5}$$

with

$$\alpha_1' = \alpha_1, \ \beta_1' = \beta_1, \ \gamma_1' = \gamma_1,$$

$$\alpha_n' = \alpha_n, \ \beta_n' = \beta_n - \frac{\delta_n}{\gamma_{n-1}'}\alpha_{n-1}', \ \gamma_n' = \gamma_n - \frac{\delta_n}{\gamma_{n-1}'}\beta_{n-1}', \ \delta_n' = 0, \quad n \geq 2.$$

and construct a continued fraction to compute the QNM spectrum

$$\frac{a_1}{a_0} = -\cfrac{\gamma_1'}{\beta_1' - \cfrac{\alpha_1'\gamma_2'}{\beta_2' - \cfrac{\alpha_2'\gamma_3'}{\beta_3' - \cdots}}} \tag{A6}$$

where $a_0$ can be set to 1, and $a_1 = -\alpha_0'/\beta_0'$ from (A5).

---

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
