# Peer review of "Quasi-normal modes of microscopic Black Holes in non-perturbative quantum gravity"

_SciPost Physics_

## Round 1 · Referee Report · Anonymous (Referee 3) · 2025-9-16

The referee discloses that the following generative AI tools have been used in the preparation of this report:
I used ChatGPT 1.2025.245 for spell check of my text
Report
Based on both content and presentation, I do not consider the manuscript suitable for publication in SciPost. In my view, the work is more appropriate for a more specialized journal.
The scope of the paper is rather narrow and technical, while some of the underlying assumptions appear highly ad hoc and speculative, restricting the potential readership. A central issue concerns the choice of mass profile. Although I acknowledge the difficulty of deriving an ab initio solution for μ, equations (21) and (25) appear entirely arbitrary. The brief justification for the 1/r dependence is insufficient, and without mapping the mass-expansion coefficients to a physical model, the QNM calculations and stability analysis risk becoming formal exercises. According to the journal’s guidelines, submissions should “provide a novel and synergetic link between different research areas.” Here, I find it difficult to identify such a connection. From the explanations and assumptions in the text, it is unclear in which physical scenario this analysis could yield genuinely new results.
The presentation also requires significant improvement. The paper contains numerous typos and inconsistencies that hinder readability. For instance, multipolar indices are sometimes omitted, the scalar field is denoted alternately by \phi and \varphi, and certain quantities (such as the solid angle and the QNM frequency \omega, which is also used to label a function acting on the box operator) are left undefined. Even if trivial, all definitions should be included for the sake of self-consistency. In addition, the language is occasionally too informal (e.g., phrases like “Even though this mess”). The manuscript would have benefitted from thorough proofreading prior to submission.
Several aspects of the framing are confusing. The title and text repeatedly refer to “QNMs of microscopic black holes,” yet the analysis concerns scalar field excitations on a fixed background, not gravitational perturbations. Strictly speaking, this is not a study of black hole QNMs in the usual sense.
I also note a number of problematic statements:
-“For the linear perturbations, one can safely disregard a field back-reaction.”
This is misleading. To me this is true for a free scalar with a stress-energy tensor quadratic in the field and without gravitational coupling. In general, whether backreaction can be neglected depends on the field content, its couplings, and the structure of the equations.
-“One of the most important classes of BH perturbations is the so-called quasi-normal modes (QNMs).”
This is highly inaccurate. QNMs are not a class of perturbations but rather characterize the damped oscillatory response of a black hole to perturbations, after the prompt signal.
-“to observe QNMs due to incredible improvements in observational techniques in recent times”
This claim is debatable. Detecting QNMs is distinct from disentangling corrections relative to GR. Current studies suggest that the latter remains unlikely, particularly for EFT-motivated gravity scenarios (see e.g. arXiv:2505.23895).
In summary, the manuscript suffers from unclear assumptions, limited physical justification, and serious presentation issues. I cannot recommend publication in SciPost in its current form.
Recommendation
Reject
Strengths
- The object of study is well defined.
- The numeric method is appropriate.
Weaknesses
- The manuscript contains a lot of wrong or misleading statements. The overall style is inappropriate, having many colloquial expressions and subjective qualifiers.
- The motivation for the complex coordinate dependent mass as a model for non-perturbative quantum gravity effects is poor.
- The correctness of some of the results is questionable.
Report
The manuscript is devoted to a study of the quasinormal modes of a test scalar field with complex mass in the Schwarzschild background. The motivation of introducing the effective complex mass for the black-hole perturbations as a result of the UV completion for gravity seems doubtful. The correction to the propagator of a test field changes the dispersion relation for that field and does not affect the gravity side since the backreaction is not taken into account. The background remains Schwarzschild, i.e. equally describes micro black holes as well as supermassive black holes. Moreover, the form of the propagator suggests that the effective mass should be a function of the energy ω rather than a function or the radial coordinate. Thus, I do not see how the choice of mass (25) corresponds to the consideration in Sec. II.
In my opinion, the style of the manuscript is not appropriate for a scientific paper. The subjective terms such as "surely", "way more complicated", "this mess", "this is not the thing which should bother us" or "nice exponential" etc. should be avoided.
There are also many wrong or misleading statements about quasinormal modes:
- The following paragraph in the introduction is wrong:
"One of the most important classes of BH perturbations is the so-called quasi-normal modes (QNMs), which correspond to the dumped outgoing GWs which a perturbed BH emits at a ring-down stage. Obviously, the lower the dumping factor is the more long-range effect such perturbations have. Thus it is therefore very tantalizing to find out such modes in a spectrum of perturbations. Given the importance and especially our current abilities to observe QNMs due to incredible improvements in observational techniques in the recent times [9–13], QNMs is a subject of very intensive current study in different settings [14–47] as well as references therein and multiple other references."
1) QNMs are not classes of perturbations. 2) QNMs do not correspond to the damped outgoing GWs. 3) Ringdown stage of the GW signal does not emit anything. 4) QNMs are the spectrum of the characteristic oscillations, not a subset of perturbations spectrum of some kind. 5) The damping rate (not a factor) of QNMs is not related to the range, but rather to the characteristic time of the correponding oscillation.
- Sec. IIIA starts with
"It is obvious from (11) that for Im(ω) > 0, the perturbation will grow with respect to time. However, such modes in a standard case can be excluded by the argument first presented in [105] which we briefly review here."
This is a misleading statement. The modes with Im(ω) > 0 grow by definition. They are not excluded by certain arguments. In [105] it is proved that they don't exist for certain kind of the effective potentials.
Eq. (15) reflects the conservation of energy and is correct for any kind of the effective potential, not only if V(r) is real. Thus, the last paragraph of Sec. IIIA is wrong.
- The following sentence in Sec. IV
"Even though we introduce an ad hoc potential it will allow us to see viability of our computations for different dependence of a scalar field mass on the radius"
is also misleading.
The authors use the well known effective potential for a massive scalar field. Ad hoc term is the scalar field mass, which is complex and depends on the radial coordinate.
Additionally, the correctness of the continued fraction method as described in the appendix is questionable.
-
It is unclear how the authors fix the sign of χ in (A1), which is defined as a square root of a complex value.
-
The prefector in (A2) is correct only when b=0. For b≠0 there is an additional subdominant term in the wave function asymptotic.
-
The infinite continued fraction (A6) converges fast only for small mass. When the parameter a is large, the continued fraction converges slowly and requires a Nollert improvement (see e.g. [8]).
Requested changes
- Address the criticisism in the report.
- Add a discussion on why the field mass depends on the radial coordinate and why this dependence is given by inverse powers of r.
- Add a discussion convergence test of the continued fraction method for nonsmall values of mass.
Recommendation
Ask for major revision
Strengths
1- The work gives a comprehensive analysis of the Quasi-normal mode spection of microscopic black holes in the presence of a complex mass.
2- The analysis is well-motivated from the perspective of quantum gravity.
Weaknesses
1- The description on how to transit from the model defined in eq. (26) to the quasi-normal modes given in Figs. 3, 4, 5, could be more detailed in order to ease the reproducibility of the results.
Report
Requested changes
1- page 1, 3rd paragraph. Please double check whether "dumping factors" should read "damping factors".
2- page 1, end of 3rd paragraph. The comment "and multiple other references" is not precise. If they are relevant for the present work, they should be cited explicitly. If they are irrelevant, the comment can be removed.
3- kindly add equation numbers to all equations in the manuscript.
4- eq. (16): please explain the subscript $t$ on $\Psi_t$ and $G_t$. The notation is not explained.
5- page 11, last paragraph: Please check if "tales" should read "tails".
6- Please double-check the explanation given between eqs. (18) and (19). The exposition on the singularities is not clear and should be improved.
7- please consider enlarging the captions in the figures. In their present size, they are very difficult to read.
8- It would be worthwhile to add some more details on the implementation of the partial fraction method in Appendix A. At which $n$ is the recursion truncated. A comment on which order is sufficient to arrive at the numerical results reported in the main section would also be helpful.
Recommendation
Ask for minor revision
Dear Editor and Referee, below we copy the latex content of an attached pdf file in which all the latex formatting is processed.
Dear Editor and Referee,
First of all, we would like to thank you for the attention to our paper and valuable comments and suggestions.
Below we put our answers to the points raised by the Referee.
Point~1 is indeed a misprint which we will correct.
Point~2 --- we will replace a vague sentence "and multiple other references." by "(for a recent review see, for instance [2505.23895])" and will insert a corresonding reference.
Point~3 --- we agree that adding numbers to equations will simplify orienting in the manuscript. All the omitted numbers will be added. Also missing commas and full stops will be added after equations for clarity.
Point~4 --- Notations after equation (16) (which WILL BECOME equation (20) after renumbering)
$G_t$ means $\partial_t G$ and this will be clarified.
Point~5 is indeed a misprint which will be corrected.
Point~6 --- Explanation between equations (18) and (19) (which WILL BE equations (22) and (23) after renumbering)
We will add the following more explicit phrase after mentioning the asymptotic behavior:
This functional dependence introduces two branch points in the complex $\omega$-plane at $\omega=\pm\mu$ (or $\pm a$ in our notation), which generate the branch cut singularities of the Green’s function.
Point~7 --- Figures will be improved by enlarging corresponding fonts.
Point~8 --- Extra clarifications regarding the convergence.
By a convergence test at some chosen data points, we use continued fraction with depth 100 so that the relative error is less than $10^{-5}$.
We will introduce this extra note in the text as well add more details of the method used to make a reader aware of them.
Sincerely,
the authors

Author: Alexey Koshelev on 2025-09-22 [id 5845]
(in reply to Report 2 on 2025-08-21)Dear Editor and Referee,
First of all, we would like to thank you for the attention to our paper and valuable comments and suggestions.
Below we put our answers to the points raised by the Referee.
As a general note, we will parse the text carefully to remove ambiguous or inappropriate for a research paper style words and phrases.
Point~1 A paragraph from the introduction ``One of the most important classes of BH perturbations is the so-called quasi-normal modes (QNMs), which correspond to the dumped outgoing GWs which a perturbed BH emits at a ring-down stage. Obviously, the lower the dumping factor is the more long-range effect such perturbations have. Thus it is therefore very tantalizing to find out such modes in a spectrum of perturbations. Given the importance and especially our current abilities to observe QNMs due to incredible improvements in observational techniques in the recent times [9–13], QNMs is a subject of very intensive current study in different settings [14–47] as well as references therein and multiple other references.'' 1) QNMs are not classes of perturbations. 2) QNMs do not correspond to the damped outgoing GWs. 3) Ringdown stage of the GW signal does not emit anything. 4) QNMs are the spectrum of the characteristic oscillations, not a subset of perturbations spectrum of some kind. 5) The damping rate (not a factor) of QNMs is not related to the range, but rather to the characteristic time of the correponding oscillation.
Our answer:
We agree with the referee that some statements in this paragraph can be expressed in a more accurate way. We propose to replace it with the following text. Quasinormal modes (QNMs) are the characteristic oscillations of black holes, stars emerging as a result of their perturbation. These modes have complex-valued frequencies for which an imaginary part is related to the damping rate. QNMs are solutions to the perturbed field equations with purely outgoing boundary conditions at spatial infinity and purely ingoing boundary conditions at the horizon. Given the possibility to observe QNMs of astrophysical black holes in gravitational wave signals from the black hole mergers [9–13], QNMs are a subject of very intensive current study in different settings, see [14–47], as well as references therein.
Point~2.1 Sec. IIIA starts with "It is obvious from (11) that for $Im(\omega) > 0$, the perturbation will grow with respect to time. However, such modes in a standard case can be excluded by the argument first presented in [105] which we briefly review here." This is a misleading statement. The modes with $Im(\omega) > 0$ grow by definition. They are not excluded by certain arguments. In [105] it is proved that they don't exist for certain kind of the effective potentials.
Our answer:
We agree that this can be phrased better and we will put it as follows in the revised version: It is obvious from (11) that for $Im(\omega) > 0$, the perturbation will grow with respect to time as they should in this case. However, such modes in many studied situations, such as BH solutions in GR, for example, can be shown not to appear in the spectrum. The corresponding consideration was first presented in [105] which we briefly review here. It will appear crucial that the potential $V(r)$ is real-valued in those situations.
Point~2.2 Eq. (15) reflects the conservation of energy and is correct for any kind of the effective potential, not only if V(r) is real. Thus, the last paragraph of Sec. IIIA is wrong.
Our answer:
Here we have to emphasize that the last term in equation (14) being added up to its complex conjugate will in fact not be a total time derivative if $V(r)$ is complex-valued. Therefore, equation (15) will not be restored and its analog will not play a role of the energy conservation. Namely, we will get $V\Psi\partial_t\Psi^+V^\Psi^*\partial_t\Psi$ which is not a total time derivative if $V$ is not real. Thus, essentially the suggested steps from (13) to (15) do not provide a time conserved quantity as along as the potential is complex-valued. However, this is enough to claim that QNM growing in time can appear as solutions to equations and require extra conditions to be imposed in order to avoid them. We will introduce this extra explanation including extra formulae in the revised version of the manuscript to make this crucial point clear to readers. In particular we will introduce another notation $\tilde V(r)$ for the potential in formulae (13)-(15) to underline that this is a local notations for this part of the text.
Point~3 The following sentence in Sec. IV "Even though we introduce an ad hoc potential it will allow us to see viability of our computations for different dependence of a scalar field mass on the radius" is also misleading.
Our answer:
This sentence will be replaced by the following extended comment: "Even though this potential is introduced ad hoc it captures the main distinguishing properties of gravity theories with higher and infinite derivatives. There are two important points to reiterate here. First, as was shown in Section II higher derivative models in general may lead to new kind of excitations which characterized by complex masses. This is also valid for gravity models. Second, we recall, that in the case of gravity we do not expect any extra excitations at all around Minkowski background and we can correspondingly adjust the parameters of a model (see (2) and a formula after it). This implies that any possible new excitations must effectively disappear in the asymptotically flat regime (for $r\to\infty$) and this corresponds to the fact that characteristic complex masses of new excitations should vanish for growing $r$. Formula (4) suggests that masses will be characterized by the Riemann tensor components which in case of the Schwarzschild background behave as inverse powers of the radius. We thus proceed with the following form of the potential:"
Moreover we will extend 2nd paragraph on page 3 in Introduction. --We will emphasize more the following logic: From the point of view of QFT a quantum gravity theory should be first defined around a flat (Minkowski) space-time. However, known approaches to non-perturbative quantum gravity end up with the presence of functions of the d'Alembertian operator in the quantum effective action. Such operators are known to generate extra states around non-flat space-times. These states are named Background Induced States and they feature complex masses squared. --We will add: A mention that explicit computations around a (anti-)de Sitter background which demonstrate appearance of BIS-s [2405.095270]. --We will also comment on related results in [2505.00761]. The latter paper shows an appearance of similar effect in an interacting theory in de Sitter space at one loop. It shows up, for example, in a late time behavior of the two point function which cannot be effectively described by the states of real mass while it can be fitted by the complex mass. --We will add corresponding references [2405.09527, 2505.00761]. --Corresponding explanations will be integrated in the present text.
Furthermore we will insert more relevant details at the end of page 5. Namely, technical explanation of how new states appear around de Sitter background as described in 2405.09527. It is a simplest maximally symmetric space-time beyond Minkowski one where all the computations can be performed explicitly.
Further clarifications before equation (3) will be added. These clarifications will be mainly aimed at explaining an appearance of new complex mass states in other including spherically symmetric backgrounds in more details.
Last Point --- Correctness of the continued fraction method and extra clarifications
Our answer
Regarding the definition of $\chi$ in Eq.~(A1) we will state explicitly that
We hope that our answers clarify the points raised by the Referee. We will implement the above changes once we will be given an opportunity to revise our manuscript.
Sincerely, the authors

---

## Editorial Decision

unknown